# Study of the Effect of Aerosol Vertical Profile on Microphysical Properties Using GRASP Code with Sun/Sky Photometer and Multiwavelength Lidar Measurements

**Francisco Molero \*****, Manuel Pujadas and Begoña Artíñano**

Department of Environment, Centro de Investigaciones Energéticas, Medioambientales y Tecnológicas (CIEMAT), Avda Complutense, 40, 28040 Madrid, Spain; manuel.pujadas@ciemat.es (M.P.); b.artinano@ciemat.es (B.A.)

**\*** Correspondence: f.Molero@ciemat.es; Tel.: +34-913466174

**Abstract:** In this paper, we study the effect of the vertical distribution of aerosols on the inversion process to obtain microphysical properties of aerosols. The GRASP code is used to retrieve the aerosol size distribution from two different schemes. Firstly, only sun/sky photometer measurements of aerosol optical depth and sky radiances are used as input to the retrieval code, and then, both this information and the range-corrected signals from an advanced lidar system are provided to the code. Measurements taken at the Madrid EARLINET station, complemented with those from the nearby AERONET station, have been analyzed for the 2016–2019 time range. The effect found of the measured vertical profile on the inversion is a shift to smaller radius of the fine mode with average differences of $0.05 \pm 0.02$ μm, without noticeable effects for the coarse mode radius. This coarse mode is sometimes split into two modes, related to large AOD or elevated aerosol-rich layers. The first scheme´s retrieved size distributions are also compared with those provided by AERONET, observing the unusual persistence of a large mode centered at 5 μm. These changes in the size distributions affect slightly the radiative forcing calculated also by the GRASP code. A stronger forcing, dependent on the AOD, is observed in the second scheme. The shift in the fine mode and the effect on the radiative forcing indicate the importance of considering the vertical profile of aerosols on the retrieval of microphysical properties by remote sensing.

**Keywords:** aerosols; LIDAR; sunphotometer; size distribution; GRASP; radiative forcing

## 1. Introduction

Atmospheric aerosols have a significant impact on the radiative energy budget of the Earth–atmosphere system [1]. Current radiative forcing estimates present a high level of uncertainty due to the high variability of tropospheric aerosols and limitations in the vertical characterization of several aerosol properties [2]. A large collection of methods have been developed for monitoring atmospheric aerosols in order to estimate these impacts at a global scale. Among others, remote sensing methods, both active and passive, have proven to be productive. Satellite remote sensing is the most promising way to collect information about global aerosol distributions [3]. Recent developments in spectral, viewing and polarization capabilities allow the extraction of aerosol properties from many new satellite sensors (e.g., MISR, PARASOL, CALIPSO, OMI or ATSR). They provide useful information on the spatial and temporal distribution of aerosols, especially over regions where ground monitoring is sparse (developing countries) or not available (over remote areas including many ocean regions). However, the satellite data still lack adequate vertical characterization capabilities or the

required representativeness needed to assess aerosol temporal and spatial variability due to their long overpass times.

Ground-based instruments generally allow more accurate observations of aerosol properties [4,5] but are only representative of the local area near the observation site. The organization of identical ground-based instruments into observational networks, with standardized data processing procedures, extends such data into larger geographical scales. For instance, ground-based networks, such as AERONET (Aerosol Robotic NETwork) [6], use passive ground-based remote sensing instruments based on solar radiation measurements (sunphotometers) to monitor aerosol properties worldwide. Multiwavelength measurements of direct beam sun irradiance, complemented with angular radiance measurements (almucantar measurements, a circle on the celestial sphere parallel to the horizon with constant zenith angle equal to solar zenith angle (SZA)) are routinely inverted to retrieve the microphysical and optical properties of aerosols for the entire atmospheric column function, such as aerosol optical depth (AOD), refractive indices, single scattering albedo (SSA), asymmetry factor and volume size distribution (VSD) [5,7]. The AERONET code has been tested and upgraded for more than two decades, and it has been implemented successfully in the worldwide network of instruments, providing useful information for satellite validation [8,9] and climate studies, where the constraining of aerosol properties has proven relevant to restrict certain scenarios [10].

Despite recent efforts to locate lidar instruments beside AERONET stations, the present reality is that information on aerosol vertical profiles is currently lacking for the vast majority of AERONET locations. Therefore, the inversion algorithm needs to assume a vertical distribution of aerosol, either a Gaussian profile or an exponential decay constrained to the measured AOD, in order to estimate microphysical properties [11]. The effect of aerosol vertical variability on sky radiance ground measurements was studied by Dubovik and King [5], concluding that it is rather modest in comparison with effects caused by aerosol VSD variability and it can often be neglected. In order to reduce the inherent uncertainty produced by the assumption of a standard profile, almucantar sky radiances are primarily selected in the inversion process, rather than principal plane measurements (where the azimuth angle of observations is kept constant and equal to the solar azimuth angle, and measurements are extended to the full celestial sphere). Atmospheric layers are therefore viewed with similar geometry in observations with sky radiances in the solar almucantar configuration, reducing the sensitivity to aerosol vertical variations, at least in single-scattering approximation [5].

However, more complex atmospheric situations can jeopardize this assumption. For instance, aerosol properties may change vertically when upper layers of long- or medium-range transported aerosols such as desert dust or those produced by forest fires are present. This will introduce an unknown error in the retrieved variables. In order to take into account the aerosol vertical distribution, the lidar (light detection and ranging) technique represents a powerful tool because of its capability to provide high-resolution aerosol profiles. Multiwavelength lidars can provide a better characterization of the aerosols by taking advantage of the additional information provided by the wavelength dependence of the backscatter and extinction coefficients. This allows for a more detailed discrimination of aerosol types [12,13]. There are several networks of lidar instruments, either worldwide, such as MPLNET (Micro-Pulse Lidar NETwork) [14], or at the continental scale, such as EARLINET (European Aerosol Research LIdar NETwork) [15], which offer a good spatial coverage of optical aerosol properties with high vertical resolution. Moreover, most EARLINET and MPLNET sites are co-located with AERONET sites, adding complementary vertically resolved aerosol backscatter or extinction profiles to the column-integrated optical properties. EARLINET is formed by 31 multiwavelength Raman lidar stations, including the one located in Madrid that is employed in this study. It started in 2000 and nowadays it is integrated into the ACTRIS (Aerosols, Clouds, and Trace gases Research InfraStructure; www.actris.eu) European project, along with several other instrument networks.

Several inversion techniques have been developed along EARLINET/ACTRIS multiple projects' lifetimes, most of them focused on retrieving vertically resolved microphysical properties from the combination of lidar-provided vertical profiles and sun/sky photometer data. Among them,

LIRIC (Lidar Radiometer Inversion Code [16]) can be mentioned, which requires the AERONET column-integrated retrievals and produces vertically resolved fine and coarse mode aerosol volume concentration by incorporating the multiwavelength backscattering lidar signals. A later development is the GARRLiC (Generalized Aerosol Retrieval from Radiometer and Lidar Combined data; [17]), which uses as inputs the almucantar sky radiances, direct sun optical depth and multiwavelength range-corrected signals in order to derive the vertically resolved aerosol microphysical and optical properties. The AERONET code and its several updates during the last few decades have influenced the above-mentioned codes; for instance, the modification of the AERONET code to incorporate information from the PARASOL instrument was used in the early stages of the GARRLiC development as a guide to include lidar profiles into the retrieval code. The development of inversion techniques capable of combining information from sunphotometer and multiwavelength lidar is an active research line nowadays.

A recently developed algorithm is the Generalized Retrieval of Aerosol and Surface Properties (GRASP, www.grasp-open.com; [18]) code. It evolves from GARRLiC and the heritage of the AERONET inversion scheme and it is being developed with the aim to allow multiple types of information to be considered in the retrievals, including multi-pixel multi-angle satellite data, column-integrated information and vertical profiles. It also allows contributions from the community, as it is offered as open-source code within the gitlab platform. Recent studies have used the GRASP code to retrieve aerosol properties by means of satellite images [19], polar nephelometer data [20], spectral AODs from sun/sky photometers [21–23], a combination of polarized and non-polarized sky radiances along with the direct sun AODs [24] and, finally, the information from sky camera images [25]. In this work, only the single-pixel capabilities of the GRASP code will be studied in order to retrieve the volume size distribution of aerosols by the combination of the AODs, sky radiances and lidar profiles, taking advantage of the GARRLiC scheme available in the GRASP code [17,26].

The main objective of this work is the assessment of the effect on the aerosol VSDs retrieved by the GRASP inversion code when combining sunphotometer and multiwavelength lidar measurements and compare the output with those obtained using sunphotometer measurements only. We combined the measurements of the ground-based lidar from the EARLINET station and the sun–sky photometer from AERONET, both instruments co-located in Madrid (Spain). This paper is structured as follows: Section 2 describes the experimental site along with the details of the instrumentation employed and the methodology to retrieve the aerosol properties by means of the GRASP code; Section 3 discusses the main results found regarding the comparison of the obtained aerosol side distributions and, finally, the main conclusions are summarized in Section 4.

## 2. Instrumentation and Methods

### 2.1. Experimental Site

The experimental site at CIEMAT (Centro de Investigaciones Energéticas, Medioambientales y Tecnológicas, 40.457°N, 3.726°W, 663 m asl) is located in the northwest of the city of Madrid, which is located in the center of the Iberian Peninsula (See Figure 1). The site is considered as urban background due to the close proximity of a park (dehesa de la villa) and the fair distance of main traffic avenues. The Madrid metropolitan area has a population of nearly 6 million inhabitants and the number of vehicles total almost 3 million. This produces a typically urban atmosphere, mainly influenced by road traffic emissions, small industrial activity and domestic and institutional heating in winter. Other contributions to the Madrid pollution, taking into account that the closest large city, Barcelona, is nearly 600 km away, can be reduced to long-range transport episodes, such as mineral dust events. Saharan dust intrusions have been established that can significantly affect aerosol concentrations measured in the Madrid region in certain meteorological situations [27]. On the contrary, the cleansing effect on the Madrid atmosphere is generally linked with the arrival of Atlantic air masses, producing a significant reduction in the particulate matter levels [28]. Simultaneous ground-based remote

sensing measurements were carried out at the site with the following instruments: the vertically resolved aerosol profile was provided by a multiwavelength advanced lidar system (Madrid-CIEMAT ACTRIS station) and the column-integrated optical properties were derived from sky and direct sun measurements provided by an automatic photometer (AEMET-AERONET station).

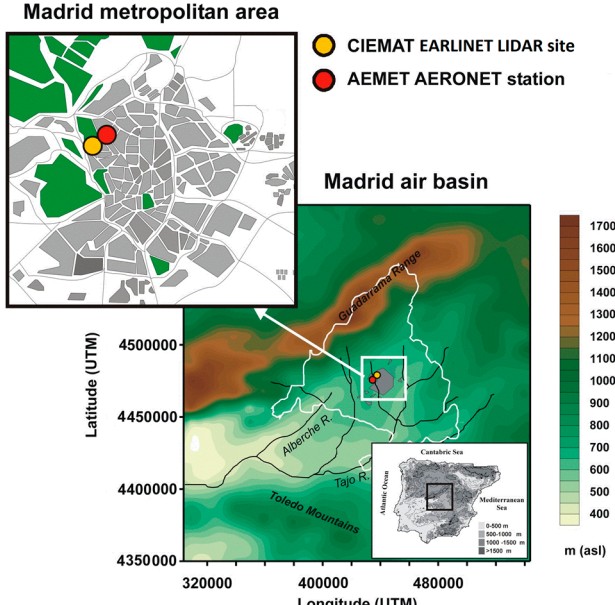

**Figure 1.** Geographical location of the EARLINET lidar site (yellow point) and the Madrid AERONET station (red point) with respect to the metropolitan area.

### 2.2. Instrumentation

#### 2.2.1. Lidar Measurements

The multiwavelength lidar instrument is a non-commercial modular instrument designed and built at the CIEMAT facilities. It is schematically composed of a transmitter sending three vertical Nd:YAG laser beams at wavelengths of 355, 532, and 1064 nm, with energies larger than 50 mJ/pulse for the three wavelengths and a 30 Hz repetition rate. The receiving line consists of a 30-cm diameter Newtonian telescope, equipped with a wavelength separation unit using dichroic mirrors and interferential filters to split the collected radiation into five channels. Three elastic signals at 355, 532, and 1064 nm and two Raman channels at 387 and 607 nm (nitrogen Raman-shifted signal from 355 and 532 nm, respectively) are detected. However, the signal-to-noise ratio for the Raman channels is very low at daytime, so these channels were not used in this study. The bi-axial configuration of the system produces full overlap between the laser beam and the telescope field of view at around 300 m above the exit window. Further description of the system can be found in [13,29]. The lidar instrument has been in regular operation since May 2006 within the EARLINET network. Lidar signals are recorded with 1-min resolution (1800 laser pulses), but later, 30-minute files are averaged in order to obtain range-corrected signals (RCS) with signal-to-noise ratio value larger than 3 up to the tropopause. The height range of lidar measurements is limited due to partial overlap in the near range and low SNR in the far range, so the actual lidar profiles used in the present study have been limited to an altitude range from 0.5 to 10 km. Below this range, the range-corrected signal is considered constant with the value of the lowest full-overlap measured point, while above, it is assumed exponentially decreasing from the highest obtained value to zero at the top of the atmosphere. Further processing includes background noise subtraction and altitude correction. Finally, a logarithmical range scale with 60 points between minimum and maximum altitudes is used as in Lopatin et al. [17]. Such decimation of lidar signals in

logarithmic scale over altitude provides useful noise suppression due to the exponential decrease in air density with altitude.

The lidar-derived extinction coefficient profile can be integrated to obtain the AOD at the lidar wavelength. The AOD of each layer can be calculated as the height-integrated extinction coefficient computed from:

$$\text{AOD}(\lambda) = \int_{z_1}^{z_2} \alpha_{\text{ext}}(\lambda)\text{d}z \tag{1}$$

where z is the height above ground level, $z_1$ and $z_2$ represent the vertical bounds of an atmospheric layer and $\alpha_{\text{ext}}$ is the extinction coefficient. The total atmosphere AOD can be obtained if $z_1$ is selected at ground-level and $z_2$ at the top of the atmosphere. These values can be compared with those provided by the sun photometer for the closest ones during daytime measurements, previously converted to the lidar wavelengths by means of the Ångström relation (440–870nm).

### 2.2.2. Sun/Sky Photometer Data

A CE-318-4 sun/sky photometer (Cimel Electronique) has been operated by the Spanish meteorological agency (AEMET) since March 2012, in a nearby location (40.452°N, 3.724°W, 680 m asl) located 500 m to the east of the lidar station. This instrument is integrated in the AERONET network and further description of the calibration, processing and standardization of these instruments can be found in Holben [6]. The microphysical properties of the aerosols, such as single-scattering albedo, phase function, aerosol optical depth, and the VSDs, can be derived from spectral sky radiance measurements at 440, 675, 870, and 1020 nm, performed at the almucantar plane, and solar direct irradiance measurements at the same wavelengths. The selected wavelengths lack significant gas absorption and present the highest sensitivity to the typical sizes of the main types of aerosols [30]. The sky radiance has been averaged between the two almucantar branches (clockwise and counterclockwise scans), discarding differences larger than 20% and limiting the azimuth angle range to 3.5°–178°, similarly to the AERONET version 2 level 1.5 criteria [31]. Sky radiance data have been convoluted using a square filter of 10 nm width centered at the photometer effective wavelengths, using the "2000 ASTM Standard Extraterrestrial Spectrum Reference E-490-00" (http://rredc.nrel.gov/solar/spectra/am0) to normalize the resulting spectra. Level 1.5 AERONET data have been downloaded, selecting version 3. This data level has been chosen as it is better than level 2.0 data, which involve further quality assurance criteria due to their larger availability, since level 2 requires an AOD @ 440 nm greater than 0.4 to exclude more uncertain aerosol absorption estimates, and such high AODs are scarcely observed over Madrid. Other criteria applied to level 2 data (sky residual error, SZA > 50°, minimum number of measurements) have been imposed on the level 1.5 data to avoid unreliable cases, and a visual inspection of the complete dataset was performed to ensure that only physically representative cases were included in the analysis. Several studies have followed the same strategy [32,33], highlighting the usefulness of this data level, which usually allows a better exploitation of the AERONET database for locations with smaller AODs all year round. Since almucantar measurements are performed only for SZA > 50°, a further limitation is that the lidar measurements must be performed within ±60 min of almucantar measurements in order to ensure reliable temporal correspondence between both datasets. Finally, instrument uncertainties are established according to AERONET standards [6], considering the AOD @ 440nm uncertainty within ±0.01, while the almucantar radiances bear a larger uncertainty of ±5%.

### 2.3. GRASP Inversion Code

The Generalized Retrieval of Aerosol and Surface Properties (GRASP) code (http://www.grasp-open.com/, [18]) is being developed as an open-source algorithm with the versatility to retrieve optical and microphysical aerosol properties from multiple data inputs, such as satellite images, polar nephelometer, sun/sky photometer, sky camera images, and lidar data. Rigorous inversion of the various combinations of these data inputs is achieved by means of multi-term least square method (LSM) [34]. This fitting algorithm can be statistically optimized to allow a flexible retrieval based

on the diverse observations. Since the present study focuses on a single location, the multipixel inversion will not be used in this work. As was mentioned in Section 1, the GARRLiC algorithm was incorporated into the GRASP code by Lopatin et al. [17], allowing the retrieval of microphysical properties of aerosols from sky/sun photometers and lidar data. The column-integrated aerosol VSDs are retrieved by adjusting 22 logarithmically equidistant triangle bins from 0.05 to 15 μm radius similarly as the AERONET code performs such calculations. Another approach suggested in some studies is the approximation of the VSDs by bimodal log-normal distributions, generally described by six parameters (radius, spread and volume of each of the fine and coarse mode), reducing the information required by the binned VSDs. A problem arises when the retrieved VSD is not perfectly log-normal, as some VSDs present asymmetrical mode shapes, and in other cases, some VSD retrievals have a pronounced trimodal structure [35]. In these circumstances, a strategy based upon simplified bimodal VSDs would not provide correct retrievals; therefore, the former strategy is preferred despite more cumbersome calculations.

The complex refractive index for both modes was assumed the same due to the limited information content of radiometric observation. The differentiation and retrieval of both the VSDs and the complex refractive indices for each mode from remote sensing is a very challenging task [36]. Sensitivity studies by Dubovik et al. [7] found that the retrieval of bi-component aerosol was not unique. As a result of this feature, the operational AERONET algorithm uses the assumption of mono-component aerosol with size-independent complex refractive index. Similar tests were performed using the Madrid database, obtaining non-convergent or unusual VSDs; therefore, a mono-component aerosol, assigning the same complex refractive index for both modes, was chosen. In general, random simulated error tests show that the uncertainty in the VSD parameters increases as the aerosol load decreases [21]. The Bidirectional Reflectance Distribution Function (BRDF) is used to take into account the part of measured sky radiance that has its source in the light reflected by the Earth's surface. GRASP uses the Li–Ross model [37,38]. The albedo values are obtained from AERONET data.

Figure 2 shows the two different schemes employed. In the first one (marked in green), the information from the sunphotometer is fed into the GRASP algorithms, similar to the AERONET procedure. In the second scheme (green + red options), both the sunphotometer information and the lidar profiles are fed into the GRASP algorithm. Figure 2a shows the AERONET Direct-Sun AOD @ 1020 nm (red dots) and the AERONET Dubovik Level 1.5 Inversion AOD @ 1020 nm (red squares), which are calculated only on those measurements that fulfill the level 2 requirements mentioned above (symmetric almucantar, more than 7 angles, SZA > 50°). During each day, sunphotometers measure around forty direct sun measurements (red dots) in fair weather situations, without clouds that limit the measurements. Sky radiance measurements (red open squares) are more limited, reaching only around eight measurements per day suitable for the retrieval of microphysical properties by means of the inversion code. The AERONET AOD errors were assumed constant and equal to 0.01 and only shown in the dark squares for clarity. Red dots have the same error bars. These values are compared with the AOD @ 1064 nm (dark red squares) calculated by integrating the lidar-derived profile of the extinction coefficient, mentioned above in Equation (1), for the complete lidar signal, reaching more than 15 km, although it has been truncated in the figure once no further aerosol layers were observed aloft (Figure 2c). The error is calculated from the error propagation of the lidar signals. EARLINET stations perform regular measurements on Mondays, at 14 h UTC (all time values are in UTC in figures and text) and after sunset, and on Thursdays after sunset in order to study aerosols from a climatological point of view. Therefore, only Monday measurements at daytime coincide temporally with AERONET measurements. As can be seen, the lidar measurements, at 14 h UTC, normally occur before the first AERONET data can be inverted and microphysical properties derived, usually after 15 h UTC, and only those closest to 60 min are analyzed. Figure 2d shows the VSDs obtained when the first scheme is employed, the green VSD when the second is employed, and the red VSD of Figure 2d and the comparison with that provided by AERONET (black curve in Figure 2d).

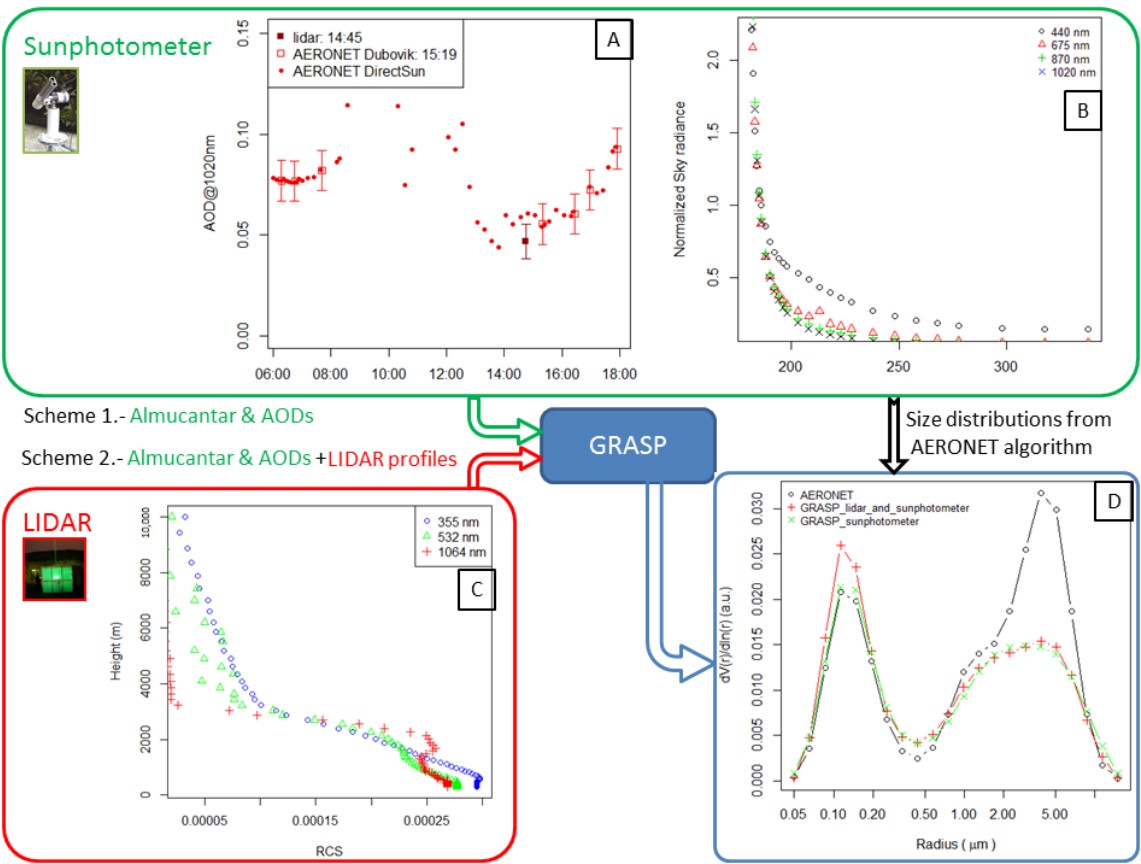

**Figure 2.** Schematic representation of the two procedures. (**a**,**b**) show the products provided by the sunphotometer (AOD @ 1020 nm values and almucantar sky radiances) and (**c**) the vertical range-corrected lidar profiles in GRASP format. The retrieved VSDs are shown in (**d**).

## 3. Results and Discussion

The Madrid site database has been explored for suitable cases between January 2016 and December 2019. EARLINET and AERONET databases were searched for coincident measurements within one hour. In total, 116 AERONET inversions (almucantar and AOD in cloud-free conditions) were found with suitable lidar profiles. The average temporal difference between lidar and sunphotometer measurements was 16 ± 11 minutes, with a maximum separation of 55 min and some cases coincident in the same minute. Figure 3 shows the AOD @ 440 nm, Angstrom Exponent between 440 and 870 nm (AE440-870), and single scattering albedo at 440 nm for all the analyzed measurements, covering four years. As can be seen, AOD in Madrid is usually low, with an average AOD @ 440 nm of 0.12 ± 0.09, with occasional larger values related to Saharan dust intrusions. Only one value above 0.4 was measured, explaining the necessity of employing level 1.5 data and preventing the use of level 2.0. The mean AE440-870 is 1.25 ± 0.36, indicating small aerosols produced by urban pollution, with occasional reductions down to 0.25 in coincidence with long-range transport of Saharan dust to the site. Finally, the SSA @ 440 nm has a mean value of 0.87 ± 0.17, expected for urban aerosols, with reduced absorption.

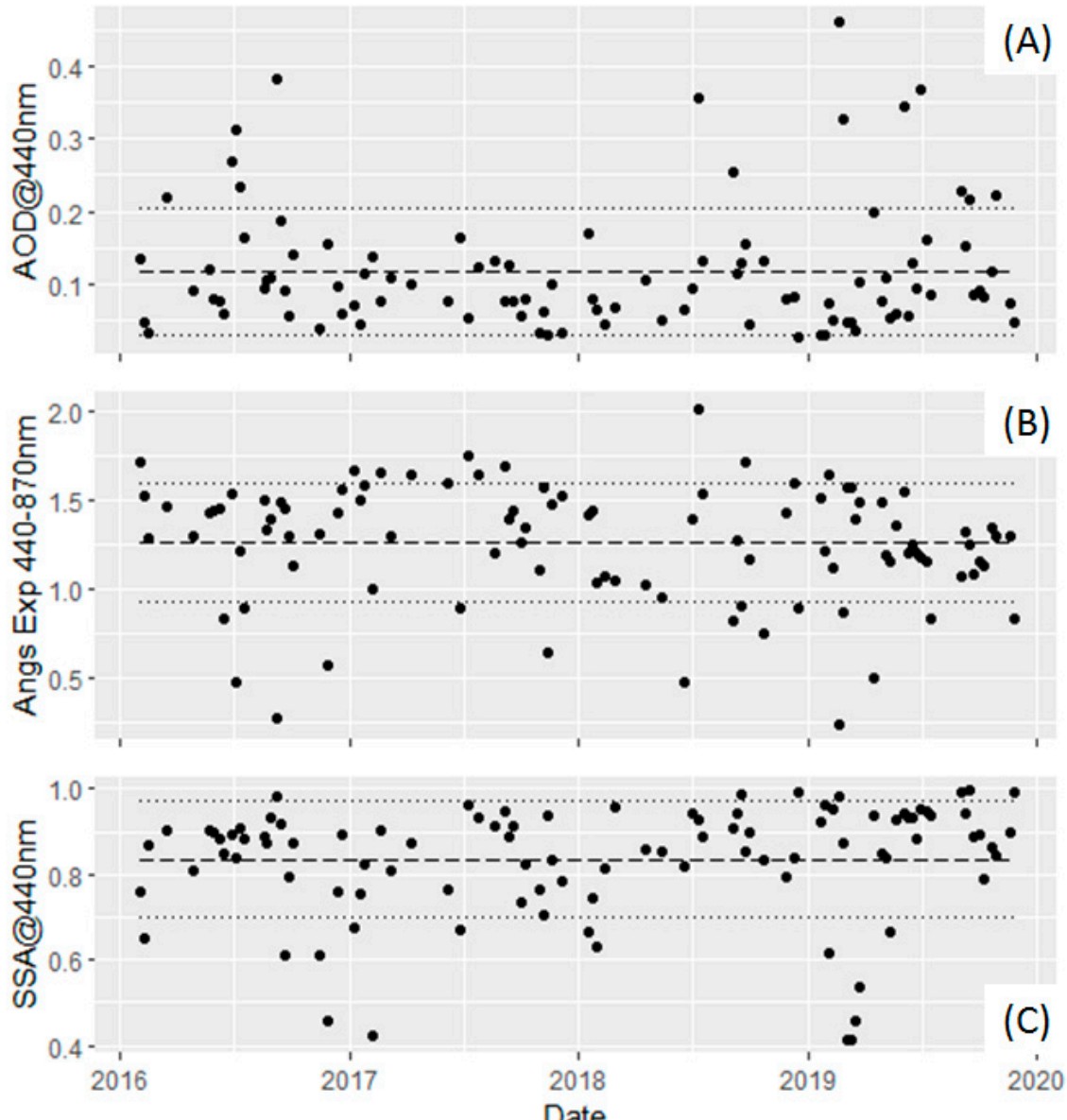

**Figure 3.** Temporal evolution of those AERONET measurements in coincidence with EARLINET profiles (within 1 h) of (**A**) AOD at 440 nm, (**B**) Angstrom Exponent between 440 and 870 nm, and (**C**) single scattering albedo for each measurement time of the study (2016–2019).

Figure 4 shows three examples of the VSDs obtained in different situations, in order to explain some of the features observed before proceeding to the statistical analysis of all the measurements. In general, the column-integrated VSDs provided by the inversion code are typically bimodal, with the first modal diameter between <0.1 and 0.4 μm and the second between 0.5 and >10 μm, with a minimum point of around 0.4 μm. In Figure 4a, a high AOD situation with an aerosol layer reaching 3 km in a well-mixed layer is presented. In this case, both schemes produce similar VSDs, with a shift to a smaller radius in the fine mode when the aerosols' vertical profiles are inserted into the GRASP code (scheme 2). The fine mode shows a shift in the center radius from 0.2 μm for the sunphotometer scheme only (scheme 1) to 0.13 μm when the lidar information is included in the inversion code. This behavior is systematically observed in the results and it affects the statistics of all cases shown in Figure 5. The coarse mode also shows a radius shift and a wider mode, but this behavior changes among the different retrievals. The AERONET VSD shows better agreement with the scheme 2 VSD for the fine mode—again, a frequently observed result among the cases analyzed.

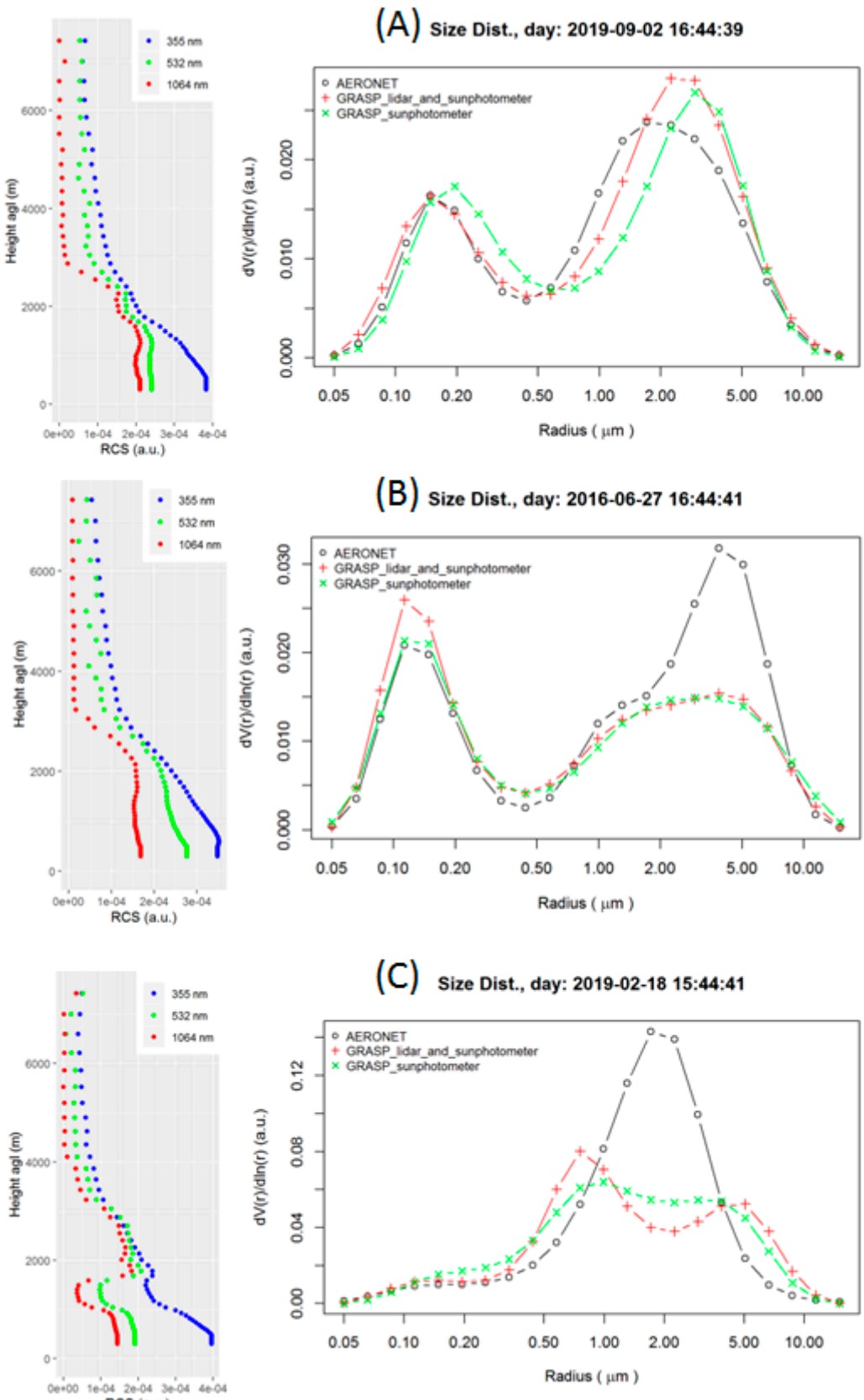

**Figure 4.** Three selected cases showing (**A**) good agreement among the VSDs, (**B**) discrepancy in the coarse mode with AERONET VSD, and (**C**) dust-rich elevated layer affecting the VSD (bottom panel). Left panels correspond to lidar profiles, right panels are the VSDs.

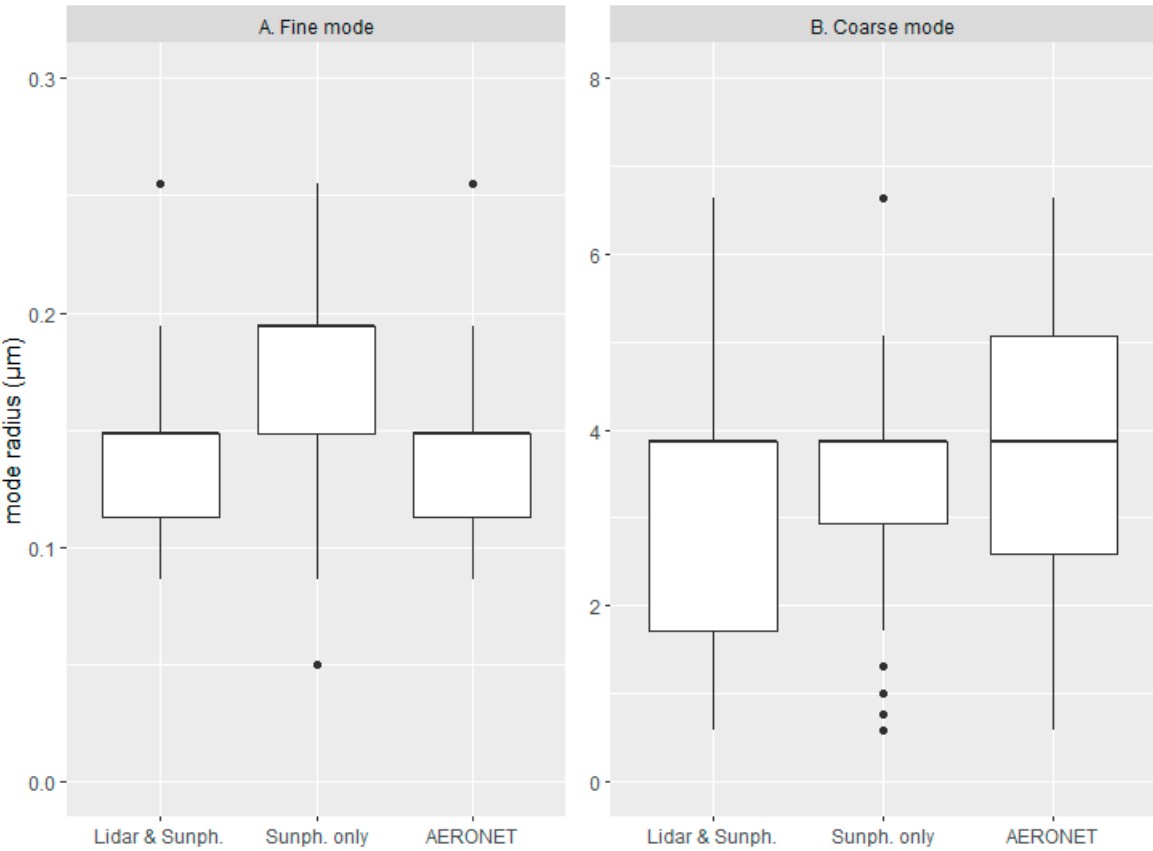

**Figure 5.** Boxplots of the mode radius for (**A**) fine and (**B**) coarse modes for the two different GRASP schemes (Scheme 1, labeled as Sunph. only. Scheme 2, labeled as Lidar & Sunph.) and AERONET.

Figure 4b shows a case of low AOD and a well-mixed aerosol layer 3 km high, where the two schemes produce similar results, with a small increase in the fine mode volume for the lidar plus sunphotometer scheme and a very similar coarse mode for both schemes. A discrepancy is observed in the AERONET VSD, showing a two-lobes coarse mode, with a strong lobe centered at 5 μm and a smaller one at 1.5 μm. The GRASP code produces a wide coarse mode covering the same radius range as the AERONET two lobes. There are several cases that show this feature, indicating a tendency of the AERONET code towards a larger coarse radius, as is shown in Figure 5.

Finally, a case with aloft dust-rich layers between 1.5 and 3 km from a Saharan dust intrusion is shown in Figure 4c. In this case, the VSDs are strongly affected by the coarse dust particles, producing a very small fine mode, similar for the three schemes, and a larger coarse mode that changes among the schemes. For scheme 1, when only sunphotometer data is used in the GRASP code, a two-lobes coarse mode is obtained, centered at 1 and 5 μm. The inclusion of the vertical profile in the inversion produces a stronger peak at 0.8 μm, better differentiated from the peak at 5 μm. Unusually, in this case, the AERONET VSD does not produce a peak at 5 μm but a strong peak at 2 μm.

The VSDs obtained in this last case present larger uncertainties because the vertical profile comprises two types of aerosol: the usual local aerosol up to 1 km and Saharan dust between 1.5 and 3 km. With the schemes selected for this work, no practical distinction between these two types can be obtained, as both modes share a common refractive index. Better results can be obtained using a bi-component scheme because the Saharan dust mainly comprises the coarse mode. However, such approaches present convergence problems, as was mentioned in Section 2.3; therefore, further developments are required to tackle this problem.

Figure 5 shows the boxplots for the radius of the fine (left panel) and coarse (right panel) modes for all the measurements. The boxplots show the mean value at the top of the quartile boxes due to

the low resolution of the radius, since the fine mode is represented by 9 bins and the coarse mode by 13 bins. The statistical values are shown in Table 1. The above-mentioned effect of a shift in the fine mode radius is observed, with a mean radius value equal to 0.189 ± 0.021 µm for the first scheme and 0.146 ± 0.025 µm for the second. Unusually, the AERONET fine mode radius, 0.154 ± 0.027 µm is closer to the second scheme, indicating that the AERONET inversion code differs from GRASP, since both codes use the same input information. Regarding the coarse mode, both schemes show similar mean values (3.532 ± 1.708 µm for scheme 1 and 3.189 ± 1.467 µm for scheme 2), although the second scheme tends to produce a lower radius, down to 1.7 for the first quartile, and the first scheme only produces a radius down to 2.94 µm in the same quartile. The AERONET coarse mode radius shows larger values, with mean equal 3.732 ± 1.545 µm. This could be due to the observed tendency of the AERONET code to produce a strong mode at 5 µm, which is not replicated by any of the GRASP schemes.

**Table 1.** Statistical values of radius for the fine and coarse modes of the different schemes.

| Mode | Scheme | Min | 1st Q | Median | Mean | 3rd Q | Max |
|---|---|---|---|---|---|---|---|
| | Lidar and sunph. | 0.086 | 0.113 | 0.148 | 0.146 | 0.148 | 0.439 |
| Fine | Sunphotometer only | 0.05 | 0.148 | 0.194 | 0.189 | 0.194 | 0.439 |
| | AERONET | 0.086 | 0.113 | 0.148 | 0.154 | 0.148 | 0.439 |
| | Lidar and sunph. | 0.576 | 1.708 | 3.857 | 3.189 | 3.857 | 6.64 |
| Coarse | Sunphotometer only | 0.576 | 2.94 | 3.857 | 3.532 | 3.857 | 15.0 |
| | AERONET | 0.572 | 2.59 | 3.857 | 3.732 | 5.061 | 6.64 |

The volume of the modes is represented as boxplots in Figure 6 and summarized in Table 2. The larger variability of this parameter, related with the AOD, prevents a clear observation of the effect among the different schemes. Both GRASP schemes produce the same median mode volume for the fine and coarse modes, with only slightly larger values for the sunphotometer scheme. The AERONET operational code produces a smaller median for both modes. The mean values for this parameter present some problems due to its variability, with a very large mean value for the second scheme fine mode, compared with that for the other schemes, due to several very large values at some of the inversions.

In order to evaluate this feature, the volume of both modes for the three different schemes is represented versus AOD @ 440 nm in Figure 7. The variability of the different cases studied prevents a clear visualization of the tendency, but the expected increase in volume with larger AOD is observed for both modes. The fine mode shows different linear fits for the different inversion schemes, with the larger volumes and also the steepest slope (0.017) for scheme 1, in good agreement with the boxplot shown in Figure 6. When the vertical profiles are included (scheme 2), the slope is reduced to 0.014. The AERONET VSDs are smaller, with a smaller dependency of the AOD (0,013). The coarse mode shows very similar tendencies for the two GRASP schemes (0.646 and 0.650), indicating that the introduction of the vertical information does not affect the volume of the coarse mode. The AERONET coarse mode shows the steepest slope (0.827), probably influenced by the cases with larger AOD, which produces different coarse mode shapes, as shown in Figure 4c.

**Table 2.** Statistical values of volume for the fine and coarse modes of the different schemes.

| Mode | Scheme | Min | 1st Q | Median | Mean | 3rd Q | Max |
|---|---|---|---|---|---|---|---|
| | Lidar and sunph. | $6.21 \times 10^{-4}$ | $1.35 \times 10^{-3}$ | $2.09 \times 10^{-3}$ | $6.66 \times 10^{-2}$ | $3.28 \times 10^{-3}$ | 5.06 |
| Fine | Sunphotometer only | $5.76 \times 10^{-4}$ | $1.6 \times 10^{-3}$ | $2.39 \times 10^{-3}$ | $3.69 \times 10^{-3}$ | $4.06 \times 10^{-3}$ | $4.06 \times 10^{-3}$ |
| | AERONET | $4.58 \times 10^{-4}$ | $9.53 \times 10^{-4}$ | $1.39 \times 10^{-3}$ | $2.63 \times 10^{-3}$ | $2.27 \times 10^{-3}$ | $2.63 \times 10^{-3}$ |
| | Lidar and sunph. | $5.44 \times 10^{-4}$ | $3.71 \times 10^{-2}$ | $6.52 \times 10^{-2}$ | $8.98 \times 10^{-2}$ | $1.08 \times 10^{-1}$ | $4.39 \times 10^{-1}$ |
| Coarse | Sunphotometer only | $4.62 \times 10^{-4}$ | $3.77 \times 10^{-2}$ | $6.69 \times 10^{-2}$ | $8.62 \times 10^{-2}$ | $1.1 \times 10^{-1}$ | $4.65 \times 10^{-1}$ |
| | AERONET | $3.56 \times 10^{-4}$ | $3.22 \times 10^{-2}$ | $5.49 \times 10^{-2}$ | $8.57 \times 10^{-2}$ | $1.07 \times 10^{-1}$ | $4.72 \times 10^{-1}$ |

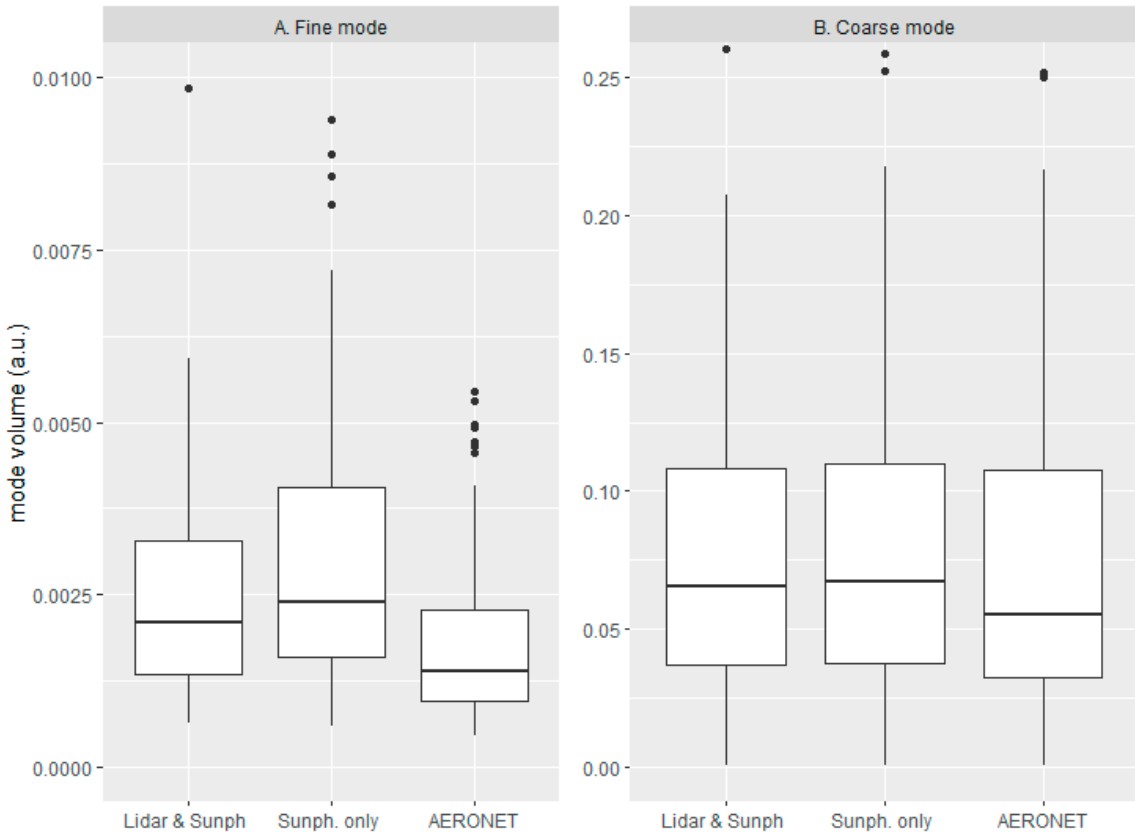

**Figure 6.** Boxplots of the volume for (**A**) fine and (**B**) coarse modes for the two different GRASP schemes (Scheme 1, labeled as Sunph. only. Scheme 2, labeled as Lidar & Sunph.) and AERONET.

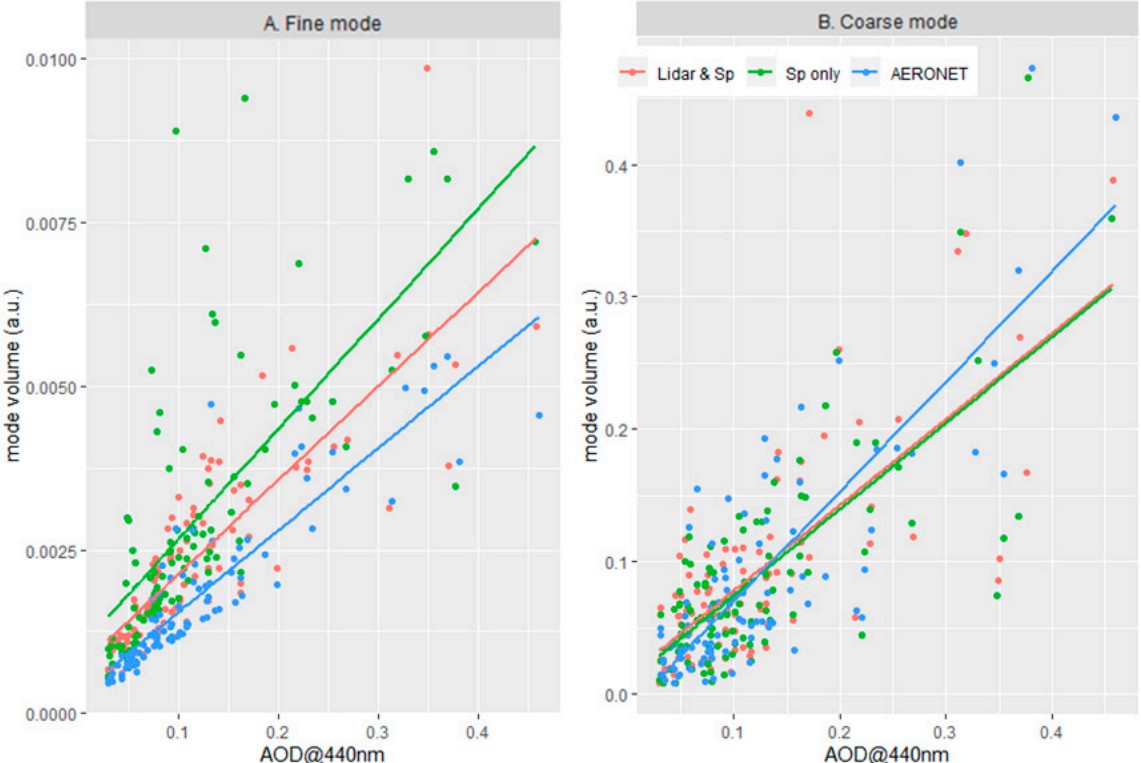

**Figure 7.** Volume area versus AOD @ 440 nm for the (**A**) fine and (**B**) coarse modes of the different schemes and their linear fit.

The parameters of the linear fits are summarized in Table 3. As can be seen in the last column of the table, the R-squared values of all the fits are low, with larger values for the AERONET (0.798 for the fine mode and 0.672 for the coarse) and lower values for the two GRASP schemes, explaining between 0.519 and 0.693 of the fine mode variability and between 0.486 and 0.549 of the coarse mode. The low values are attributed to the natural variability of the atmospheric situations during the four years of the study and they will be studied in more detail in further investigations.

**Table 3.** Linear fit parameters and R-squared values for volume area vs. AOD @ 440 nm corresponding to the fine and coarse modes of the different schemes.

| Mode | Scheme | a | b | $R^2$ |
|---|---|---|---|---|
| Fine | Lidar and sunphotometer | $6.95 \times 10^{-4}$ | 0.014 | 0.693 |
| | Sunphotometer only | $9.78 \times 10^{-4}$ | 0.017 | 0.519 |
| | AERONET | $2.93 \times 10^{-4}$ | 0.013 | 0.798 |
| Coarse | Lidar and sunphotometer | $1.39 \times 10^{-2}$ | 0.646 | 0.486 |
| | Sunphotometer only | $9.88 \times 10^{-3}$ | 0.650 | 0.549 |
| | AERONET | $-1.20 \times 10^{-2}$ | 0.827 | 0.672 |

Finally, the different VSDs may affect the atmospheric radiances measured either at ground level or from space. This radiative forcing can be calculated with the GRASP code by solving the radiative transfer equations, considering a plane parallel approximation using the retrieved aerosol properties. Figure 8 shows the radiative forcing values obtained versus the AOD, along with the tendencies provided by linear fit for the bottom of the atmosphere (BOA, left panel) and top of atmosphere (TOA, right panel) for the different schemes. Similar behavior is observed for the BOA and TOA, with close tendencies for the two GRASP schemes but slightly stronger for the lidar and sunphotometer scheme, with a tendency of $-46.4$ for BOA and $-38.6$ for TOA; with respect to the sunphotometer only scheme, tendencies of $-43.2$ for BOA and $-36.4$ for TOA are found. Regarding the AERONET values, a larger difference is observed for the BOA ($-122.5$), but this could be due to the different method of calculation of the radiative forcing employed by AERONET, which considers the upward flux to be equal for the cases with aerosols and without them. Such an assumption is applicable at the TOA but not at the BOA, possibly explaining the larger values observed at BOA for the AERONET scheme. For the TOA, all radiative forcing equations are equal but, still, some difference is observed in the AERONET scheme ($-66.5$ respect to $-38.6$ or $-36.4$). This could be due to the differences mentioned above in the retrieved VSDs.

The parameters of the linear fits are summarized in Table 4. In this case, the R-squared values, shown in the last column of the table, account for a significant fraction of the variability of the data, with values as large as 0.943 for the radiative forcing at TOA calculated with the sunphotometer only scheme. In general, this scheme produces the larger R-squared values (0.942 for TOA and 0.926 for BOA), with still significant values for the lidar and sunphotometer scheme (0.872 for TOA and 0.889 for BOA) and lower values for AERONET (0.652 for TOA and 0.651 for BOA). In summary, a small effect on the radiative forcing is observed between the GRASP schemes, indicating the robustness of the sunphotometer only scheme for this application, in accordance with the conclusions reached by AERONET studies [7], but larger discrepancies are observed with the AERONET scheme, which again requires future investigation.

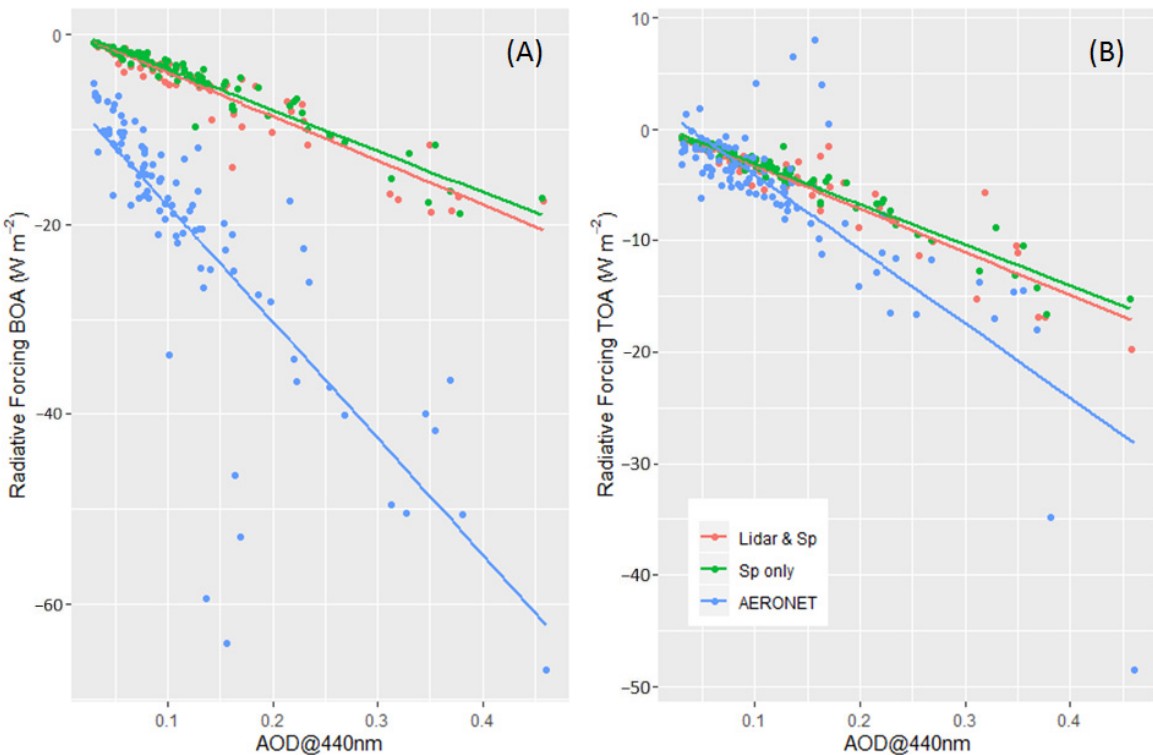

**Figure 8.** Radiative forcing versus AOD @ 440 nm for the (**A**) BOA and (**B**) TOA of the different schemes and their linear fit.

**Table 4.** Linear fit parameters and R-squared values for radiative forcing vs. AOD @ 440 nm corresponding to the fine and coarse modes of the different schemes.

| Level | Scheme | a | b | $R^2$ |
|---|---|---|---|---|
| | Lidar and sunphotometer | 0.6525 | −46.4 | 0.889 |
| BOA | Sunphotometer only | 0.688 | −43.2 | 0.926 |
| | AERONET | −5.768 | −122.5 | 0.651 |
| | Lidar and sunphotometer | 0.556 | −38.6 | 0.872 |
| TOA | Sunphotometer only | 0.511 | −36.4 | 0.943 |
| | AERONET | 2.509 | −66.5 | 0.652 |

## 4. Conclusions

In this work, we have explored the use of coincident observations of sun/sky photometer and lidar measurements in the GRASP code to retrieve column-integrated microphysical aerosol properties such as volume size distributions. The main goal of the present work was to analyze the effect of adding information about the aerosol vertical profile to the retrieval of VSDs from spectral optical depth measurements. This has been accomplished by analyzing the results provided by two different schemes: firstly, only sun/sky photometer measurements of aerosol optical depth (AOD) and sky radiances are used as input to the retrieval code, and secondly, both this information and the range-corrected signals from an advanced lidar system are provided to the code. Measurements taken at the Madrid EARLINET station, complemented by those from the nearby AERONET station, have been analyzed for the 2016–2019 time range. As a relevant result, it can be mentioned that the main effect of the measured vertical profile on the inversion is a shift to a smaller radius of the fine mode, with average differences of 0.05 ± 0.02 μm, without noticeable effects for the coarse mode. This coarse mode is sometimes split into two modes, related to a large AOD or elevated aerosol-rich layers. The results were also compared with those provided by AERONET, observing an unusual persistence of a large

mode centered at 5 μm. This change in the size distributions affects slightly the radiative forcing calculated also by the GRASP code. A stronger forcing, dependent on the AOD, has been observed in the second scheme, indicating that the aerosol vertical information provided by lidar is relevant in the inversion algorithm of aerosol optical properties such as the VSD, a parameter required in aerosol radiative forcing estimations. The shift in the fine mode and the effect on the radiative forcing indicate the importance of considering the vertical profile of aerosols in the retrieval of microphysical properties by remote sensing, with observable effects in the modal radius and also in the radiative forcing calculations.

Several studies are planned for the future. The large variability observed in the data may be addressed in follow-up studies with a better characterization of atmospheric situations, especially clean versus polluted conditions and cases with aerosol-rich aloft layers. Regarding the GRASP code, further efforts are planned to retrieve reliable products using a bi-component scheme, as the possibility to distinguish indices of the refraction of fine and coarse particles may improve the characterization of aerosols and the different effects observed in this study.

**Author Contributions:** Conceptualization, F.M., M.P. and B.A.; Data curation, F.M.; Formal analysis, F.M., M.P. and B.A.; Funding acquisition, B.A.; Investigation, F.M., M.P. and B.A.; Methodology, F.M., M.P. and B.A.; Project administration, B.A.; Validation, F.M.; Visualization, F.M.; Writing—original draft, F.M.; Writing—review and editing, M.P. and B.A. All authors have read and agreed to the published version of the manuscript.

**Funding:** This research was funded by European Union's Horizon 2020 research and innovation programme through project ACTRIS-2 (grant 654109), the Spanish Ministry of Economy and Competitivity (CRISOL, CGL2017-85344-R and ACTRIS-ESPAÑA, CGL2017-90884-REDT) and Madrid Regional Government (TIGAS-CM, Y2018/EMT-5177).

**Acknowledgments:** We thank AERONET and Juan Ramón Moreta González, of AEMET, for their effort in establishing and maintaining the Madrid site. We would like to acknowledge the use of GRASP inversion algorithm software (http://www.grasp-open.com) in this work.

**Conflicts of Interest:** The authors declare no conflict of interest.

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
