# Peer review of "Study of the Effect of Aerosol Vertical Profile on Microphysical Properties Using GRASP Code with Sun/Sky Photometer and Multiwavelength Lidar Measurements"

_remotesensing, doi:10.3390/rs12244072_

Round 1

Reviewer 1 Report

This is a nice paper and can be accepted after some minor revisions. 1. There are no “take-home -messages” in the abstract. Include your important findings here. 2. Include some recent relevant studies in introduction. 3. The Section 2 “2. Instrumentation and Methods” is too long and unfocused. Concise it at least by 50%. What level data you have used? Mention it clearly. 4. You must provide the figures in high resolution. This applies to all figures. 5. Conclusion: make it more integrated and compact. Good luck.

Author Response

Response to Reviewer 1 Comments

Point 1: There are no “take-home -messages” in the abstract. Include your important findings here. 

Response 1: The abstract has been revised to highlight the important findings, rewriting them in a more “take-home-messages” style

Point 2: Include some recent relevant studies in introduction

Response 2: Recent studies, some of them from this same year, have been added to the introduction

Point 3: The Section 2 “2. Instrumentation and Methods” is too long and unfocused. Concise it at least by 50%. What level data you have used? Mention it clearly.

Response 3: Section 2.2 has been rewritten with the aim to reduce its lengths and improve the focus. Regarding data levels, the study has used AERONET level 1.5 data, but filtered following Level 2 quality assurance criteria, except the AOD@440nm > 0.4 requirement that AERONET apply. If this requirement is applied to Madrid data, the resulting database is too scarce. Regarding lidar data, EARLINET network classify all cases employed in this study as Level 2, and they are available in the EARLINET database. But in order to input the lidar profile into GRASP, some reformatting is necessary, so raw data (level 1) was used

Point 4: You must provide the figures in high resolution. This applies to all figures

Response 4: All figures have been revised and improved in resolution.

Point 5: Conclusion: make it more integrated and compact

Response 5: The conclusion section has been revised, highlighting the relevant findings and trying to make it more integrated and compact

Reviewer 2 Report

The Study of the effect of aerosol vertical profile on microphysical properties using GRASP code with sun/sky photometer and multiwavelength lidar measurements provides useful information about the improvement in code.

I have some questions ?

1. Why GRASP giving higher values in small size fraction while AERONET gives very high values at coarse size fraction. While it seems are in good agreement.

 2. You applied GRASP CODE to lidar and sun photometer, did you tried other way around ? Mean did you applied AERONET CODE to LIDAR and check how its responding. 

3. It may possible GRASP will be underestimating in large size fraction and overestimates in small size. 

4. At many places authors uses , instead of .   kindly check it, line 372 , 373 and other text as well.

Author Response

Response to Reviewer 2 Comments

Point 1: 1. Why GRASP giving higher values in small size fraction while AERONET gives very high values at coarse size fraction. While it seems are in good agreement.

Response 1: Our results show the mentioned “good agreement” only in some cases, with others showing larger disagreements. The reliability of the modes is connected with the AOD of the case. Larger AOD produce more reliable inversion products, but also those cases are normally produced by larger aerosols, contributing to the coarse mode. For instance, during Saharan dust intrusions. The fine mode seems to be more difficult to determine, especially in low AOD cases. Since AERONET code may be using some conditions that GRASP ignore, that may explain the different results.

Point 2: You applied GRASP CODE to lidar and sun photometer, did you tried other way around ? Mean did you applied AERONET CODE to LIDAR and check how its responding.

Response 2: No, we cannot apply AERONET code because it isn’t available to the public, as the GRASP code is. It would be an interesting study because some differences are observed between the GRASP code with sunphotometer input and AERONET products. This means that AERONET code is applying some conditions or restrictions that haven’t been published. 

Point 3: It may possible GRASP will be underestimating in large size fraction and overestimates in small size.

Response 3: It is difficult to establish that from our results due to the large variability of the different cases. It may be possible, of course, but a more restricted experimental design, maybe with synthetic data, would be better to validate such statement. We will keep it in mind for future studies and thanks the reviewer for the suggestion.

Point 4: At many places authors uses , instead of .   kindly check it, line 372 , 373 and other text as well.

Response 4: The words “instead of” has been changed to “rather than” or “better that” when appropriate
